# InfiMM-WebMath-40B: Advancing Multimodal Pre-Training for Enhanced Mathematical Reasoning

**Xiaotian Han**[1*]   **Yiren Jian**[1*]   **Xuefeng Hu**[1*]   **Haogeng Liu**[1,2,*]   **Yiqi Wang**[1*]
**Qihang Fan**[1,2]   **Yuang Ai**[1,2]   **Huaibo Huang**[2]   **Ran He**[2]   **Zhenheng Yang**[1]
**Quanzeng You**[1†]
[1]ByteDance, Inc    [2]Chinese Academy of Sciences

## Abstract

Pre-training on large, high-quality datasets is essential for improving the reasoning abilities of Large Language Models (LLMs), particularly in specialized fields like mathematics. However, the field of Multimodal LLMs (MLLMs) lacks a comprehensive, open-source dataset for mathematical reasoning. To fill this gap, we present InfiMM-WebMath-40B, a high-quality dataset of interleaved image-text documents. It consists of 24 million web pages, 85 million image URLs, and 40 billion text tokens, all carefully extracted and filtered from CommonCrawl. We outline our data collection and processing pipeline in detail. Models trained on InfiMM-WebMath-40B demonstrate strong performance in both text-only and multimodal settings, setting a new state-of-the-art on multimodal math benchmarks such as MathVerse and We-Math. We release our data at https://huggingface.co/datasets/Infi-MM/InfiMM-WebMath-40B.

## 1   Introduction

Recent advancements in Large Language Models (LLMs)[1, 2, 12] have improved their ability to handle complex reasoning and multi-step mathematical problems through techniques like Chain-of-Thought (CoT) prompting[54]. These models excel from basic GSM8K word problems [10] to high school-level MATH tasks [19]. Specialized smaller LLMs like DeepSeekMath-7B [49] and InternLM-Math [58] have also made notable progress in mathematics, demonstrating strong performance in focused domains.

Although most mathematical knowledge is text-based, visual elements such as figures and diagrams are essential for understanding abstract concepts. To integrate these visual components, Multimodal LLMs (MLLMs) like G-LLaVA [14], Math-LLaVA [50], and MAVIS [65] have been developed. These models enhance reasoning by incorporating visual inputs through embeddings from pre-trained models like CLIP [47] and SigLIP [61], and use multimodal instruction datasets such as Geo170k [7], MathV360K [51], and MAVIS-Instruct [66].

However, introducing new knowledge during instruction fine-tuning is challenging [69], often leading to hallucinations [16], particularly due to limitations in dataset scale and quality. While large corporations benefit from proprietary datasets, the open-source community lacks comprehensive pre-training datasets for mathematical reasoning that integrate text and visual data.

To address this gap, we introduce **InfiMM-WebMath-40B**, the first large-scale, publicly available multimodal mathematics pre-training dataset. Comprising 24 million web documents, 85 million image URLs, and 40 billion text tokens, it provides a valuable resource for training Multimodal

---

[*]Equal contributions.

[†]Corresponding author.

The 4th Workshop on Mathematical Reasoning and AI at NeurIPS'24

LLMs (MLLMs). We validate the effectiveness of InfiMM-WebMath-40B through experiments on benchmarks like MathVerse [64] and WeMath [46], showing improved performance in multimodal mathematical reasoning.

Our contributions include: (1) We introduce InfiMM-WebMath-40B, the first large-scale, multimodal math dataset for pre-training, filling a critical gap in open-source research. (2) We provide a detailed preprocessing pipeline for filtering relevant content from CommonCrawl to ensure high-quality, relevant data. (3) We demonstrate the impact of InfiMM-WebMath-40B through experiments, where our models excel on multimodal mathematical benchmarks, showcasing the dataset's potential for advancing MLLM research.

## 2   Related Work

LLMs have demonstrated potential in mathematical reasoning across various studies. To evaluate and enhance their capabilities, several math-specific benchmarks [11, 20, 18, 4, 40, 32, 67] and training datasets, both proprietary [45, 31, 27] and open-source [19, 55, 41, 53, 60], have been introduced.

The rise of Multimodal LLMs (MLLMs) has sparked interest in enhancing their multimodal reasoning capabilities. To support this, various evaluation benchmarks [62, 35, 24, 56, 38, 57, 34, 64, 46] and training datasets [7, 15, 51, 68, 26, 3, 30] have been developed to assess and enhance MLLMs' mathematical reasoning skills.

## 3   Dataset Construction

In this section, we detail the methodology used to construct InfiMM-WebMath-40B, a large-scale multimodal math dataset integrating interleaved text and image data, following approaches used in prior works [44, 29, 43]. We enhance the methodology used in the OBELICS dataset [26] by incorporating both text and corresponding image URLs.

### 3.1   Text-only Data Curation Pipeline

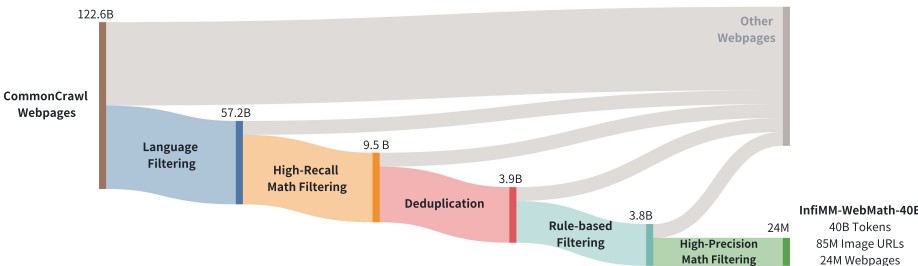

Figure 1: InfiMM-WebMath-40B data curation pipeline.

**Text Extraction and Language Filtering**   We chose Trafilatura, a Python library widely used to extract text from web pages. While effective for text extraction, Trafilatura omits mathematical symbols and equations. Therefore, the subsequent section will outline our development of a specialized extraction tool tailored for math-related content.

Following DeepSeekMath [49], we focus on retaining only Chinese and English content when constructing our dataset. To achieve this, we apply language filtering to the CommonCrawl repositories with approximately 122 billion webpages, as shown in Figure 1. For language detection, we employ a fastText language identification model [22]. This language filtering process significantly reduces the dataset size, lowering the number of pages from 122 billion to 57.2 billion.

**Mathematical Content Extraction**   Extracting mathematical content from HTML presents unique challenges, as standard tools often fail to accurately capture LaTeX equations and image URLs. After

evaluating various tools, we chose Resiliparse as the foundation for our development. Figure 2 shows a comparison of extraction results between Trafilatura and our enhanced version of Resiliparse.

**High-Recall Filtering for Mathematical Content**  Inspired by DeepSeekMath [49], we trained a fastText classifier to filter mathematical content, using half a million positive samples from OpenWeb-Math [42] and negative samples from our earlier extracted content. This filtering reduced the dataset from 57.2 billion to 9.5 billion samples, prioritizing recall with a probability threshold set at 0.4.

**Deduplication**  We applied MinHash [6] for content deduplication, following FineWeb's methodology [43]. Deduplication was performed within each snapshot and neighboring snapshot pairs, reducing the dataset by 43%, from 9.5 billion to 5.4 billion samples. URL deduplication further reduced the sample size to 3.9 billion.

**Rule-based Filtering**  We applied a few essential filtering rules, such as removing "lorem ipsum" content, applying a punctuation ratio rule for English, filtering NSFW content, and excluding documents with Unicode errors. This step eliminated 3% of the samples, resulting in 3.8B samples.

**High-Precision Filtering for Mathematical Content**  To enhance the accuracy of our labeling process, we utilized the LLaMA3-70B-Instruct model [12], using prompt formats inspired by the FineWeb-Edu dataset [33]. This approach allowed us to score the mathematical quality of each sample on a scale from 0 to 10. The full prompt is displayed in Table 3 of Appendix.

From the data remaining after rule-based filtering, we randomly sampled approximately one million entries. We assigned math quality scores and applied a threshold of 6 to select 640,000 positive samples for training our updated fastText classifier, alongside an equivalent number of 640,000 randomly selected negative samples from prior filtering steps. These positive and negative samples were combined to train the new fastText classifier.[3]

During fastText training, we applied data cleaning rules to optimize the model's performance for mathematical content (see Appendix D for details). For evaluation, we used all samples in the Geometry3K [35] benchmark as positive examples of mathematical content. With these refined preprocessing techniques, fastText's accuracy improved from 48.74% to 72.15

**Text-Only Filtering Evaluation**  We pretrained a deepseek-coder-1.3b-base model on the filtered text dataset and evaluated its performance on GSM8K [10] and the MMLU (STEM) [18]. Our model outperformed both OpenWebMath and DeepSeekMath, highlighting the quality of our dataset (results are shown in Appendix E).

## 3.2  Multimodal Data Construction

After filtering, 24 million documents with 85 million image URLs remained. We extracted image URLs from each webpage and paired them with the corresponding text, following the OBELICS format [26]. Deduplication reduced the image URLs to 23 million. Further filtering based on keyword analysis (e.g., "log", "banner", "avatar", "icon") left us with 22 million URLs, from which we successfully downloaded 14 million unique images. These images were reintegrated into the documents, resulting in 24 million records with a total of 28 million images.

## 4  Experiments

**Model Architectures**  We employ the SigLip model `siglip-so400m-patch14-384` to extract visual features, a 3-layer Perceiver Resampler [21] with 64 latents to reduce the number of tokens/features per image to 64. These visual token/feature embeddings are then concatenated with text embeddings before being fed into the LLMs (DeepSeek-Coder [17]: `deepseek-coder-1.3b-base` and `deepseek-coder-7b-v1.5`).

**Training Details**  Our training data and processes involve a three-stage approach: modality alignment, continued pre-training using InfiMM-WebMath-40B, and instruction fine-tuning. Detailed training procedures are provided in the Appendix F. We refer to our resulting model as InfiMM-Math.

---

[3]We also employ an LLM-based classifier for high-precision filtering, Appendix C shows the comparison.

Table 1: Evaluation of models on MathVerse.

| Model | Base LLM | All | Text Dominant | Text Lite | Vision Intense | Vision Dominant | Vision Only |
|-------|----------|-----|---------------|-----------|----------------|-----------------|-------------|
| Human | - | 64.9 | 71.2 | 70.9 | 61.4 | 68.3 | 66.7 |
| *Proprietary Models* | | | | | | | |
| GPT-4V | N/A | 39.4 | 54.7 | 41.4 | 34.9 | 34.4 | 31.6 |
| Gemini-Pro | N/A | 23.5 | 26.3 | 23.5 | 23.0 | 22.3 | 22.2 |
| *Open-sourced Models* | | | | | | | |
| SPHINX-Plus | LLaMA2-13B | 14.0 | 16.3 | 12.8 | 12.9 | 14.7 | 13.2 |
| G-LLaVA | LLaMA2-7B | 15.7 | 22.2 | 20.4 | 16.5 | 12.7 | 6.6 |
| InternLM-XC2 | InternLM2-7B | 16.5 | 22.3 | 17.0 | 15.7 | 16.4 | 11.0 |
| Math-LLaVA | Vicuna-13B | 19.0 | 21.2 | 19.8 | 20.2 | 17.6 | 16.4 |
| ShareGPT4V | Vicuna-13B | 17.4 | 21.8 | 20.6 | 18.6 | 16.2 | 9.7 |
| LLaVA-NeXT | LLaMA3-8B | 19.3 | 24.9 | 20.9 | 20.8 | 16.1 | 13.8 |
| LLaVA-NeXT | Qwen-1.5-110B | 24.5 | 31.7 | 24.1 | 24.0 | 22.1 | 20.7 |
| MAVIS | Mammoth2-7B | 27.5 | 41.4 | 29.1 | 27.4 | 24.9 | 14.6 |
| *Our Models* | | | | | | | |
| InfiMM-Math | DS-Coder-1.3B | 26.9 | 37.1 | 30.2 | 29.2 | 24.4 | 13.7 |
| InfiMM-Math | DS-Coder-1.5-7B | 34.5 | 46.7 | 32.4 | 38.1 | 32.4 | 15.8 |

**Evaluations on MathVerse**  In line with official MathVerse guidelines, we report the "w/o" score. The results in Table 1 show that our 7B model outperforms all open-source models, including the 110B LLaVA-NeXT, and surpasses Gemini-Pro and Qwen-VL-Max, trailing only GPT-4V. Our model demonstrates exceptional performance in the Text-Dominant, Text-Lite, Vision-Intense, and Vision-Dominant categories, highlighting its strong multimodal capabilities in processing both text and visual inputs. However, it underperforms in the Vision-Only category, likely due to limitations in our vision encoder, which processes images only at a resolution of $384 \times 384$. To validate the effect of our proposed InfiMM-WebMath-40B, we also provide ablations on the CPT and IFT datasets in Appendix G.

Table 2: Evaluations on the We-Math benchmark. AVG represents the primary metric of interest.

| Model | Base LLM | AVG ↑ | IK ↓ | IG ↑ | CM ↑ | RM ↓ |
|-------|----------|-------|------|------|------|------|
| *Proprietary Models* | | | | | | |
| Gemini-1.5-Pro | N/A | 26.4 | 42.7 | 11.2 | 20.8 | 54.8 |
| GPT-4V | N/A | 31.1 | 39.8 | 14.5 | 23.8 | 47.9 |
| *Open-sourced Models* | | | | | | |
| LLaVA-1.6 | Vicuna-7B | 3.3 | 78.3 | 2.5 | 2.1 | 89.1 |
| LLaVA-1.6 | Vicuna-13B | 5.2 | 69.1 | 3.2 | 3.6 | 86.9 |
| DeepSeek-VL | DeepSeek-7B | 6.3 | 69.1 | 4.6 | 4.0 | 84.8 |
| G-LLaVA | Vicuna-13B | 6.5 | 64.2 | 4.6 | 4.2 | 86.6 |
| Math-LLaVA | Vicuna-13B | 11.1 | - | - | - | 72.8 |
| InternLM-XC2 | InternLM2-7B | 12.7 | 56.4 | 10.5 | 7.4 | 77.6 |
| *Our Models* | | | | | | |
| InfiMM-Math | DeepSeek-Coder-1.3B | 13.1 | 56.2 | 9.1 | 9.3 | 73.7 |
| InfiMM-Math | DeepSeek-Base-7B | 20.6 | 48.8 | 12.2 | 15.2 | 61.7 |

**Evaluations on We-Math**  Here, we compare models on the We-Math benchmarks, consisting of 6.5K visual math questions. We report results on the `testmini` set using four metrics: Insufficient Knowledge (IK), Inadequate Generalization (IG), Complete Mastery (CM), and Rote Memorization (RM). As shown in Table 2, our model, InfiMM-Math, surpasses all open-source models.

## 5 Conclusions

In this work, we introduced InfiMM-WebMath-40B, the first large-scale multimodal pretraining dataset for mathematical reasoning, filling a crucial gap in open-source research. Our dataset significantly enhances models' performances on key benchmarks.

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

# A    Mathematical Content Extraction

Figure 2 illustrates a comparison of extraction results between Trafilatura and our enhanced version of Resiliparse. Our tool successfully extracts both the mathematical equations and image URLs, as highlighted in the red boxes in the screenshot from a Wikipedia webpage.

### Integral form   [ edit ]

Gauss's law may be expressed as:[6]

$$\Phi_E = \frac{Q}{\varepsilon_0}$$

where $\Phi_E$ is the electric flux through a closed surface $S$ enclosing any volume $V$, $Q$ is the total charge enclosed within $V$, and $\varepsilon_0$ is the electric constant. The electric flux $\Phi_E$ is defined as a surface integral of the electric field:

$$\Phi_E = \oiint_S \mathbf{E} \cdot \mathrm{d}\mathbf{A}$$

where $\mathbf{E}$ is the electric field, $\mathrm{d}\mathbf{A}$ is a vector representing an infinitesimal element of area of the surface,[note 2] and · represents the dot product of two vectors.

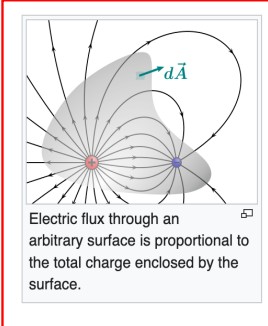

Electric flux through an arbitrary surface is proportional to the total charge enclosed by the surface.

| From trafilatura | From Ours |
|---|---|
| Integral form\n[edit]Gauss's law may be expressed as:[6]\nwhere ΦE is the electric flux through a closed surface S enclosing any volume V, Q is the total charge enclosed within V, and ε0 is the electric constant. The electric flux ΦE is defined as a surface integral of the electric field:\nwhere E is the electric field, dA is a vector representing an infinitesimal element of area of the surface,[note 2] and · represents the dot product of two vectors. | Integral form\n\n**[Image_Link]//upload.wikimedia.org/wikipedia/commons/thumb/3/32/Electric-flux-surface-example.svg/220px-Electric-flux-surface-example.svg.png [Image_Link] Electric flux through an arbitrary surface is proportional to the total charge enclosed by the surface.**\n\nGauss's law may be expressed as:\n\n$\\Phi _{E}={\\frac {Q}{\\varepsilon _{0}}}$\n\nwhere ΦE is the electric flux through a closed surface S enclosing any volume V, Q is the total charge enclosed within V, and ε0 is the electric constant. The electric flux ΦE is defined as a surface integral of the electric field:\n\n$\\Phi _{E}=$ $\\scriptstyle _{S}$ $\\mathbf {E} \\cdot \\mathrm {d} \\mathbf {A}$\n\nwhere E is the electric field, dA is a vector representing an infinitesimal element of area of the surface,[note 2] and · represents the dot product of two vectors. |

Figure 2: A comparative illustration of extraction results from a Wikipedia webpage using Trafilatura and our enhanced version of Resiliparse, highlighting the successful retrieval of mathematical equations and image URLs.

# B    Using Prompting with Llama-3-70B for Mathematical Annotation

We display the full prompt used in High-Precision Filtering for Mathematical Content in Table 3.

Table 3: Prompt for evaluating mathematical content using Llama-3-70B following FineWeb-Edu [33].

```
Below is an extract from a web page.  Evaluate the mathematical value of the extract and its potential utility as a teaching
resource in a mathematical context using the additive 10-point scoring system described below.  Points accumulate based on
the satisfaction of each criterion, with special attention to the presence and quality of mathematical equations:
- 0 points if the extract includes no mathematical content, such as only provides historical context, summarizes an article's
abstract, or exclusively features a person's resume.
- 1-2 points if the extract offers rudimentary information on mathematical subjects, even if interspersed with irrelevant
material such as advertisements or non-academic content.
- 2-4 points if the extract touches upon mathematical topics without rigorous adherence to academic standards and contains a
mix of mathematical and non-mathematical content, or if the presentation is haphazard and the writing lacks clarity.
- 4-6 points if the extract presents key concepts pertinent to educational curricula and includes mathematical equations,
albeit potentially non-comprehensive or alongside superfluous information.  It should resemble a mathematical text, such as
an introductory section of a textbook or a basic tutorial.
- 6-8 points if the extract is highly relevant to mathematics, is well-structured, and offers a clear exposition, including a
significant number of mathematical equations and solutions.  It should be akin to an in-depth textbook chapter or tutorial,
with a strong focus on mathematical content and minimal unrelated information.
- 8-10 points if the extract exhibits exceptional mathematical merit, characterized by detailed explanations, a comprehensive
array of mathematical equations, and a coherent, accessible writing style that provides profound insights into mathematical
theories and applications.
The extract:  <EXAMPLE>.
After examining the extract:  - Briefly justify your total score.  - Conclude with the score using the format:
"mathematical score:  <total points>"
```

## C  Ablation Studies on High-Precision Mathematical Content Filtering

In this section, we examine the efficacy of two classifiers—LLM-based and fastText-based—focusing on high-precision mathematical content filtering. The comparison utilizes the DeepSeek-Coder 1.3B model, which we trained on a dataset previously introduced in Sec. High-Recall Filtering for Mathematical Content with a sequence length of 4096. This model was trained to score documents based on their relevance to mathematical content on a scale from 0 to 10.

We conduct the continue pretraining of the DeepSeekCoder 1.3B model using datasets filtered by both the LLM- and fastText-based classifiers. Table 4 shows the performance results. The results highlight a length bias in the LLM-based method, which tends to favor longer documents, averaging 2,500 tokens, compared to 1,700 tokens for the FastText filter. The length bias associated with the LLM-based classifier has adversely impacted the dataset's performance on the GSM8K dataset. As indicated in the table, the LLM-filtered dataset achieved lower accuracy (17.5%) on the GSM8K dataset compared to the fastText-filtered dataset (20.2%). This decrease in performance indicates that the LLM's preference for longer documents may not align well with the requirements of datasets like GSM8K, which demand concise and precise mathematical descriptions.

Given these insights, we have decided to continue utilizing the fastText classifier for high-precision filtering in our ongoing research. Nonetheless, the implications of the LLM-based classifier require further investigation to fully understand and address its biases.

Table 4: Ablations on the high-precision filtering. The "Text Avg Length" column indicates the averaged document length after filtering by each respective classifier.

|                    | MMLU (STEM) | GSM8K | Text Avg Length |
|--------------------|-------------|-------|-----------------|
| LLM-Classifier     | 32.8        | 17.5% | 2500            |
| FastText-Classifier| 31.1        | 20.2% | 1700            |

## D  Data Cleaning Rules in FastText Training

During fastText training, we implement data cleaning rules to optimize the model's performance for mathematical content. Mathematical texts pose unique challenges due to specialized terminology, symbols, formulas, and numeric data, which differ from typical natural language and require more refined preprocessing techniques.

Our goal is to standardize and simplify the input training data while preserving essential mathematical information. Key considerations include maintaining consistency in token representation, minimizing noise from extraneous characters, and standardizing numeric values. The following steps reflect this approach:

- Utilizing the SpaCy English language model (`en_core_web_sm`), we preprocess the input text, tokenize it, and process each token by converting it to its lowercase and lemmatized form. Common placeholders are replaced, certain non-alphanumeric characters are removed, and patterns of special characters like dashes and underscores are normalized. We also strip any unnecessary whitespace, ensuring the text is well-prepared for downstream processing.
- All numeric values are replaced with the <NUM> placeholder to standardize the representation, and line breaks along with carriage returns are removed. Tokens exceeding 100 characters in English are discarded.

## E  Text-Only Filtering Evaluation

To provide a preliminary evaluation of the quality of our filtered dataset, we continue pretraining a deepseek-coder-1.3b-base model for one epoch using the filtered mathematical content in Sec. High-Precision Filtering for Mathematical Content, excluding image URLs. We validate the effectiveness of our math-related filtering with a few-shot evaluation using the GSM8K [10] and the STEM sections of the MMLU [18] benchmark.

Table 5: Evaluation of models on GSM8K and MMLU (STEM). The baseline is the deepseek-coder-1.3b-base without any training.

| Training Corpus | GSM8K | MMLU (STEM) |
|---|---|---|
| Baseline | 4.8 | 25.6 |
| OpenWebMath [41] | 11.0 | 29.6 |
| DeepSeekMath Corpus [48] | 23.8 | 33.1 |
| InfiMM-WebMath-40B (text) | 26.1 | 35.6 |

As shown in Table 5, the model trained on our InfiMM-WebMath-40B text-only dataset demonstrates competitive performance compared to OpenWebMath and the DeepSeekMath Corpus, highlighting the high quality of our dataset and the effectiveness of our filtering procedures.

## F    Training Details

**Modality Alignment Stage**    In this stage, we utilize general-purpose image-text pairs to align the visual encoder and the LLM via Perceiver Resampler. The primary objective is to minimize the domain gap between visual and linguistic modalities. To achieve this, we sample a 8 million image-text pair subset from the DFN-2B dataset [13] for the alignment training. During this stage, the vision encoder and LLM backbone are frozen, and training is focused on the Perceiver Resampler module. Training is conducted for one epoch using DeepSpeed Zero2, with the AdamW optimizer, configured with a cosine learning rate scheduler, a maximum learning rate of $1e^{-4}$, betas of $(0.9, 0.95)$, and a weight decay of 0.1.

**Continue Pre-training Stage**    We further continue pre-training our models using the InfiMM-WebMath-40B dataset to enhance the model's mathematical knowledge acquisition in a multi-modal setting. The training is conducted for one epoch using DeepSpeed Zero2, with the AdamW optimizer, configured with a cosine learning rate scheduler, a maximum learning rate of $5e^{-5}$, betas of $(0.9, 0.95)$, and a weight decay of 0.1. The context length for training examples is set to 4096, with a maximum of 32 images per example. During this stage, the visual encoder remains frozen, and training focuses on learning the Perceiver Resampler module (the visual-language connector) and the LLM.

**Instruction Fine-tuning Stage**    In this stage of training, we fine-tune our models using instruction datasets, including PGPS9K [63], Geo170k [15], TABMWP [37], ScienceQA [36], Vflan [8], Visual-WebInstruct, AI2D [25], ChartQA [38], DocVQA [39], DVQA [23], GeoQA [9], and MAVIS [66]. We find that incorporating uni-modal text instruction datasets is crucial for enhancing the models' instruction-following capabilities. Therefore, we also include pure text instruction datasets such as Math[28], MetaMathQA [59], DART-Math [52], and NuminaMath [5]. The objective of this stage is to acclimate the models to the common chat templates used in math VQA settings, thereby enabling them to better utilize the mathematical knowledge acquired in the previous stage.

We freeze the vision encoder and update the parameters of the Perceiver Resampler and LLMs. As in the previous stages, training is conducted using DeepSpeed Zero2 for one epoch, with the AdamW optimizer, configured with 2000 warmup steps, a maximum learning rate of $5e^{-6}$, betas of $(0.9, 0.95)$, a weight decay of 0.1, and cosine decay to $5e^{-7}$. The batch size is set to one per GPU, and the context length of the training examples is set to 4096. We utilize 32 A100-80G GPUs for the 1.3b models and 64 A100-80G GPUs for the 7b models.

## G    CPT and IFT Dataset Ablations on MathVerse

In this section, we conduct ablation studies on models (1) trained with and without continue pre-training (CPT), and (2) models fine-tuned on the MAVIS dataset versus a more extensive instruction fine-tuning (IFT) dataset. Specifically, we compare models trained with and without our own mathematical multi-modal pre-training dataset, InfiMM-WebMath-40B. Additionally, we evaluate two IFT dataset configurations: (a) a combination of MAVIS-Caption-to-QA, MAVIS-Existing-

Table 6: Datasets ablations (CPT and IFT) using Deepseek-coder-1.3B.

|  | CPT | IFT | MathVerse w/o score |
| --- | --- | --- | --- |
| DSC-1.3B |  | Mavis | 20.2 |
| DSC-1.3B | ✓ | Mavis | 25.1 (+4.9) |
| DSC-1.3B |  | Extended | 22.3 |
| DSC-1.3B | ✓ | Extended | 26.9 (+4.6) |

Table 7: Datasets ablations (CPT and IFT) using Deepseek-coder-1.5-7B.

|  | CPT | IFT | MathVerse w/o score |
| --- | --- | --- | --- |
| DSC-1.5-7B |  | Mavis | 22.8 |
| DSC-1.5-7B | ✓ | Mavis | 27.1 (+4.3) |
| DSC-1.5-7B |  | Extended | 23.8 |
| DSC-1.5-7B | ✓ | Extended | 29.1 (+5.3) |

Dataset-Augment, MAVIS-Caption, MAVIS-DataEngine-Geometry, and MAVIS-Meta-Question (referred to as the MAVIS dataset); and (b) a broader set consisting of the MAVIS datasets along with Vflan, VisualWebInstruct, AI2D, CHARTQA, DOCVQA, DVQA, GEOQA, DART-Math, and Numina-Math (referred to as the Extended dataset).

As shown in Table 6, in the 1.3B model, CPT improves the MathVerse scores by 4.9 and 4.6 points when IFT is performed with MAVIS and Extended datasets, respectively. Similarly, Table 7 shows that in the 7B model, CPT improves the MathVerse scores by 4.8 and 5.3 points with MAVIS and Extended datasets, respectively. In contrast, using broader IFT datasets typically enhances model performance by approximately 2 points. These results highlight the significant mathematical capabilities imparted to the models through our InfiMM-WebMath-40B for CPT.

