# OpenReview forum: "InfiMM-WebMath-40B: Advancing Multimodal Pre-Training for Enhanced Mathematical Reasoning"
_NeurIPS.cc/2024/Workshop/MATH-AI — MATH-AI 24_

### Official Review · Reviewer_bDYe · 2024-09-29
**Review of "InfiMM-WebMath-40B"**

**Rating:** 7
**Confidence:** 3

**Review:**

# Summary of paper

The paper, "InfiMM-WebMath-40B" (omitting subtitle) contributes a large scale dataset of contributed images and web document pairs. The authors describe the collection, data cleansing process, (section 3) and tunes several models with the data collected (section 4).

The paper is generally well written in clear English. However, for a dataset which is poised to be reused writ large by the Math-AI and the deep learning community more generally, there are a few items to consider to vastly improve the clarity and reusability of the contributed dataset.

My comments are mostly focused on improving clarity and increasing the future citation counts of this manuscript and the contributed data.

# Major points:

1. There should be summary statistics on the images and the size of the documents in the contributed dataset. Both by # of images and by pixels and image file formats.

2. Given that the authors use LLMs to assess quality-and subsequently threshold filter the quality score-they should also classify the type of math problem associated with the document and associated images. In "Smart Vision-Language Reasoners" presented at ICML 2024, the authors  found that the improvement on a model using images and text varied according to the math problem category (e.g. algebra vs geometry etc) therefore having problem type categories would be beneficial for ablations.

3. Given the cleansing process employed there is a likelihood of false positives and false negatives in the cleansing and filtration process. While estimating confusion matrices for each binary classifier employed may be infeasible, the top level classifier steps presented in figure 1 should have estimates for confusion matrix entries based on a small subset of randomly selected document/image pairs.

4. The math extraction tool used for extraction should be open sourced as part of the submission, no indication of whether this will be done was indicated, nor was an anonymized link to code provided. Providing code used is vital for reproducibility in science and furthering research in the Math-AI space.

# Minor points:

1. Bibliography contains multiple duplicate entries that are also not formatted correctly.

2. Authors need to clarify the process for Chinese language processsing and how the data differ for Chinese vs English, eg. summary breakdowns.

3. In appendix D the authors indicate they use a custom `<NUM>` token, to represent numeric values. Unless I misunderstand there is the potential to generate duplicates if the `<NUM>` tokens obscure distinctly different values from pages generated by web browsers in a templated manner. Was this considered and the deduplication was done before the `<NUM>` token overwrite?

3. Where will the data be made publicly available?

---

### Official Review · Reviewer_V7ES · 2024-10-07
**InfiMM-WebMath-40B: Advancing Multimodal Pre-Training for Enhanced Mathematical Reasoning**

**Rating:** 9
**Confidence:** 5

**Review:**

The paper introduces InfiMM-WebMath-40B, a large-scale, high-quality multimodal dataset designed to enhance mathematical reasoning in Large Language Models (LLMs). ​The authors detail their data collection and processing pipeline, emphasizing the integration of both text and visual data. ​

Pros
- Comprehensive Dataset: InfiMM-WebMath-40B fills a critical gap in the open-source community by providing a large-scale, high-quality multimodal dataset specifically for mathematical reasoning. ​
- Detailed Methodology: The paper offers a thorough explanation of the data collection and processing pipeline, ensuring reproducibility and transparency. ​
- Strong Experimental Results: Models trained on InfiMM-WebMath-40B achieve state-of-the-art performance on benchmarks like MathVerse and We-Math, validating the dataset's effectiveness. ​
- Innovative Filtering Techniques: The use of both high-recall and high-precision filtering methods, including the development of specialized tools for mathematical content extraction, enhances the dataset's quality. ​
- Significant Research Impact: The dataset has the potential to significantly advance research in multimodal mathematical reasoning, providing a valuable resource for the open-source community. ​

Cons
- Limited Vision-Only Performance: The models underperform in the Vision-Only category, indicating potential limitations in the vision encoder used. ​
- Resource Intensive: The model training processes are resource-intensive, which may limit accessibility for smaller research groups.

---

### Official Review · Reviewer_sSnb · 2024-10-08
**A Step Forward in Multimodal Mathematical Datasets**

**Rating:** 7
**Confidence:** 5

**Review:**

This paper introduces InfiMM-WebMath-40B, a comprehensive multimodal dataset designed for pre-training mathematical reasoning models. The authors outline the dataset construction process, evaluate its effectiveness through benchmark tasks, and demonstrate improvements over current methods. While this work represents a meaningful contribution to the field, some aspects of the presentation and methodology could be refined.

Presentation Wise :
1. The paper follows a clear and logical structure, typical of academic research.
2. The abstract concisely highlights the main contributions and findings.
3. Figures and tables are well-integrated and enhance the narrative.

Significance:
1. The introduction of a large-scale, open-source multimodal dataset specifically for mathematical reasoning is a notable advancement.
2. It addresses a critical resource gap for training multimodal language models (MLLMs) in mathematics.
3. The improved performance on benchmark tasks indicates the dataset's potential for advancing future research and practical applications.

Soundness:
1. The dataset construction methodology is meticulously described and appears rigorous.
2. The experimental setup and benchmark evaluations are appropriate and well-executed.
3. Ablation studies provide valuable insights into the impact of various components in the approach.
4. However, more detailed statistical analysis and error examination would strengthen the findings.

Strengths of the paper:
1. A comprehensive data curation pipeline that includes text extraction, language filtering, and domain-specific filtering.
2. Development of specialized tools for extracting mathematical content and equations from web sources.
3. The creation of a large-scale dataset (40 billion tokens, 85 million image URLs) combining text and visual data.
4. Demonstrated performance improvements across multiple benchmark tasks compared to existing models.
5. Thorough ablation studies that show the benefits of continued pre-training and instruction fine-tuning.

Weaknesses of the paper:
1. A lack of detailed error analysis or qualitative examples of model outputs.

Additional Comments and Questions:
1. How does the dataset address potential copyright issues related to web-scraped content?
2. What steps were taken to ensure data quality and remove harmful or biased content?
3. How does the performance of models trained on this dataset compare to human performance on the same tasks?
4. Are there plans to update or expand the dataset to keep it current?
5. How could this approach be adapted or applied to domains beyond mathematics?
6. How does the dataset handle varying mathematical notations or conventions across different sources?

This paper makes a contribution to the field of MLLMs in mathematical reasoning. The introduction of a large-scale, open-source dataset addresses a key resource gap and delivers promising results on benchmark tasks.

---

### Decision · Program_Chairs · 2024-10-08

Accept